# Slot Antennas Integrated into 3D Knitted Fabrics: 5.8 GHz and 24 GHz ISM Bands

**DOI:** 10.3390/s22072707

**Published:** 2022-04-01

**Authors:** Miroslav Cupal, Zbynek Raida

**Affiliations:** Faculty of Electrical Engineering and Communication, Brno University of Technology, 61600 Brno, Czech Republic; cupal@vut.cz

**Keywords:** textile-integrated antenna, textile-integrated waveguide, 3D knitted fabric, circular polarization, screen printing, sewing

## Abstract

In the paper, a 3D knitted fabric is used for the design of a circularly polarized textile-integrated antenna. The role of the radiating element is played by a circular slot etched into the conductive top wall of a textile-integrated waveguide. Inside the circular slot, a cross slot rotated for about 45° is etched to excite the circular polarization. The polarization of the antenna can be changed by the rotation of the cross slot. The antenna has a patch-like radiation pattern, and the gain is about 5.3 dBi. The textile-integrated feeder of the antenna is manufactured by screen printing conductive surfaces and sewing side walls with conductive threads. The antenna was developed for ISM bands 5.8 GHz and 24 GHz. The operation frequency 24 GHz is the highest frequency of operation for which the textile-integrated waveguide antenna has been manufactured.

## 1. Introduction

Thanks to the technological progress of textile-integrated electronics, more and more sophisticated devices can be integrated to the clothing [1] and textile components of flats [2], vehicles [3] and other daily use objects. Attention has now turned to textile-integrated controls and sensors improving the quality of life, health care and safety.

Textile-integrated electronics is usually designed to be wearable. In order to ensure washability and other utility properties of fabrics, only the necessary parts of electronics can be of a purely textile nature. A majority of the circuits are hidden and protected by a plastic capsule [1] or similar covers. But thanks to the rapid development of textile technologies, larger and larger portions of electronics can now become an inherent part of textile products [4].

In order to integrate antennas into textile materials, conventional planar antennas are usually redesigned for textile substrates and textile-oriented manufacturing processes. Conductive surfaces are manufactured by screen printing [5], inkjet printing [6], copper plating [7] or embroidering [8]. The conductive walls of textile-integrated structures can be sewed by conductive threads [9].

In the past years, many types of textile antennas for different applications and frequency bands were published. Dipoles [10,11,12], spiral antennas [13], patches [5,14] and planar inverted F antennas (PIFA) [15] were usually designed for ISM bands 2.4 GHz and 5.8 GHz using different fabrics and yarns. Some planar monopole antennas for UWB bands have been designed for frequencies up to 20 GHz [8]. Published papers have focused on the manufacturing processes and comparison of conventional antennas, embroidered structures and printed antennas.

A meandered asymmetric flare dipole on a polyester fabric was presented in [10]. The antenna of the size 190 mm × 25 mm operated at 500 MHz, and its performance was comparable with an antenna on a conventional microwave substrate.

The comparison of various fabric dipole antennas operating at 2.4 GHz was presented in [11]. Two embroidered antennas (a single-layer version and a double-layer one) were compared with a silver fabric antenna, which showed for about 6 dB higher gain than the embroidered ones. A circularly polarized spiral antenna designed for 420 MHz [16] belongs to the most complex antennas that have ever been published.

Comparisons of antennas manufactured from different fabrics and various conductive threads are relatively common. Nevertheless, [17] compares the patch antenna designed for 2.4 GHz, which was embroidered using different stitch directions. Due to the losses in conductive threads, the gain of embroidered patch antennas is lower than the gain of equivalent FR4 antennas. The 2.4 GHz PIFAs made from different meshes of conductive threads on various fabrics were presented in [15] and compared with the conventional metallic PIFA.

Antennas based on textile-integrated waveguides (TIW) are a special class of textile antennas. TIW is a textile version of a conventional substrate-integrated waveguide (SIW). TIW is a closed structure with the top surface and the bottom one covered by a conductive layer; side walls are sewed by a conductive thread. Since parasitic radiation of the structure is negligible, TIW components can be used to operate at higher frequencies [18].

Dealing with papers available in IEEE Xplore, the reader can find 35 papers on textile-integrated circularly polarized antennas published in the last decade. Antennas designed for 2.4 GHz ISM band were presented in 10 papers, antennas operating in 5.8 GHz ISM band were described in 12 papers, textile-integrated RFID antennas were published in 5 papers, wideband antennas for 0.3 to 3.0 GHz bandwidth were shown in 4 papers, and a GPS antenna was described in 1 paper.

Since the amount of information gathered by wearable electronics is increasing, higher and higher rates are needed to transmit data from textile-integrated objects to remote units for further processing. Therefore, textile-integrated electronics is expected to operate at higher and higher frequencies.

Dealing with the upper frequency limit of textile-integrated circularly polarized antennas, a planar monopole-like antenna for body-centric communication [19] showed the impedance bandwidth and axial-ratio one from 4 to 10 GHz, approximately. Nevertheless, the monopole was attached to the skin directly, and the tissue significantly loaded the measured antenna. When placing two similar planar textile-integrated monopoles on a common piece of fabric, a multiple-input multiple-output (MIMO) antenna was created with similar impedance and polarization properties [20].

In [21], we presented a preliminary concept of an antenna designed for the 24 GHz ISM band. The antenna consisted of a radiating slot etched to the top wall of a textile-integrated resonator. Since the radiating element was isolated from the tissue by a conductive layer, the antenna was influenced by the tissue negligibly. The antenna was manufactured by copper plating. Due by technological problems, the operation frequency of the antenna was shifted by 2 GHz. In [22], we showed technological improvements of processes to manufacture textile-integrated filters and antennas. The 24 GHz antenna was optimized for screen printing, newly fabricated, and characterized from the impedance viewpoint.

In this communication, we characterize polarization and radiation properties as well, and compare the 5.8 GHz antenna with the 24 GHz antenna to describe challenges related to the frequency upscale of the studied structure. To the best of our knowledge, such results have not been published yet. Details are given in Section 2.

In Section 3, antennas designed for ISM bands 5.8 GHz and 24 GHz are presented, and their parameters are verified by simulations and measurements. At lower ISM frequency, the available bit rate is lower and antenna dimensions are larger. For higher frequencies, the fabrication technology has to be very accurate. Considering the accuracy of knitting, screen-printing and sewing, the 24 GHz ISM band is shown to be the upper frequency limit of textile-integrated antennas, considering today’s textile technologies. According to our knowledge, a textile-integrated waveguide antenna operating at 24 GHz has not been published yet.

## 2. Materials and Methods

### 2.1. Antenna Concept

The antenna is conceived as a circular slot radiator etched into the top conductive layer of a textile integrated waveguide (TIW). In the metal surface surrounded by the circular slot, a cross slot with two perpendicular arms is placed with the longer arm touching the circular slot. The cross slot is rotated for about 45° to excite the circular polarization. Thanks to the described configuration, two modes can circulate in the circular slot and interfere with the circularly polarized wave. The whole radiator is placed half wavelength away from the end of the TIW (see Figure 1).

In Figure 1, the light–dark color represents an electrically conductive surface (a copper foil or a screen-printed layer) on the top surface of the textile substrate. Vias (stitches) connecting the top surface and the bottom one are represented by dark-gray circles. The radiating slot is depicted in white. A coaxial feed, which is fixed to the antenna from the bottom, is depicted by the dashed line.

The TIW is designed by conventional rules. For the efficient width of the TIW *w_ef_*, we use [18]:(1)$$w_{ef}=w-1.08\frac{d_{v}^{2}}{g}+0.1\frac{d_{v}^{2}}{w}$$

Here, *w* is the physical width of TIW, *d_v_* is the diameter of the thread (the metal via), and *g* is the distance between two stitched threads (metal vias) [18]:(2)$$\frac{g}{d_{v}}=2.5$$

When designing TIW, the ratio between the via diameter and the spacing of vias has to be chosen, so that the parasitic through-wall radiation is minimized. Nevertheless, the diameter of threads is given by the manufacturer, and is related to the construction or requested tension.

The role of the substrate is played by the 3D knitted fabric. The fabric is manufactured from cotton or polyester, creating two firm equidistant surfaces with normal yarns in between. Since most of the volume is filled with air, the dielectric constant is between 1.2 and 1.3. The thickness of 3D knitted fabrics can vary from 0.6 mm to 5.0 mm. Parameters of selected 3D knitted substrates are shown in Table 1.

The antennas were designed for 5.8 GHz and 24 GHz ISM bands. Both the antennas were developed following the same rules, and manufactured using the same technologies.

### 2.2. Antenna Design

The antenna radiator is etched into the top conductive layer of a textile-integrated waveguide (TIW), which is short ended and fed by a coaxial SMA probe. The feeding point is placed to the maximum of electric field in TIW. Since the electrical thickness of the textile substrate is relatively high at 24 GHz, the TIW should be fed as a common rectangular waveguide (the inner conductor is not connected to the top conductive layer of the antenna).

The center of the circular slot should be placed half wavelength far from the TIW end. The center is placed to the minimum of the electric field (to the maximum of the magnetic field) as depicted in Figure 1.

The slot radius is related to the wavelength of the radiated wave. The inner radius of the slot can be calculated as [23]:(3)$$r=\frac{c}{2\pi f\sqrt{\varepsilon_{r}}}$$

Here, *c* is the velocity of light in vacuum, *f* is frequency of operation, and *ε_r_* is relative permittivity of the textile substrate.

The width of the slot is given by the requested characteristic impedance. The cross slot consists of a longer arm and a shorter one. The length of the longer arm is given by the radius of the circular slot, and the length of the shorter one equals to:(4)$$l_{c2}=\frac{r}{4}$$

Here, *r* is the inner radius of the ring slot given by (3).

### 2.3. Technological Aspects

The technological aspects of manufacturing the mm-wave structures comprise specific features related to used materials. The manufacturing accuracy of planar microwave structures to be integrated into textile substrates is limited to ±10%. Even manufacturing repeatability and tolerances of electrical parameters of textile materials (substrates, conductive surfaces, threads) are relatively low.

In order to make the manufacturing technology with a relatively low accuracy applicable, the antenna has to be robust and insensitive to relevant dimensions. In the presented design, the radiating slot is excited by the dominant mode in the textile-integrated resonator, and field distribution of the mode is relatively stable. Deep parametric analyses proved that 10% manufacturing accuracy can be used in that specific case.

The designed antennas are manufactured with three main components, characterized as:The textile substrate. The 3D knitted fabrics are used. Parameters of fabrics were measured by the transmission line method [24], and are given in Table 1.Conductive surfaces. Screen printing was applied. Since 3D knitted fabrics are porous, the silver paste penetrates into the textile substrate and does not create a conductive surface on the textile only. Therefore, an iron-on foil is used to prevent penetration and smoothen the textile surface before printing. The thickness of the foil is 0.08 mm, and relative permittivity equals 2.1. A corresponding layer must be added to the simulation model to prevent the shift of the resonant frequency.Conductive side walls. Conductive yarns are used to sew the walls. When selecting the yarn, low resistance is requested and good mechanical properties are needed to obtain good contact between the yarn and the screen-printed surface. When manufacturing prototypes, the yarn ELITEX^®^ XT 117/f17/2ply_PA/Ag was used.

## 3. Results

### Simulations and Prototype Measurements

The 5.8 GHz version of the antenna was designed on the 3D knitted material 3D041 (*h* = 3.4 mm, *ε_r_* = 1.22), and the 24 GHz antennas were designed on 3D097 (*h* = 2.6 mm and *ε_r_* = 1.22). Dimensions of antennas are given in Table 2.

Virtual prototypes were modeled in CST Microwave Studio using the transient solver. Since the structure of knitted materials influence antenna properties negligibly due to relatively low operation frequencies [25], textile substrates were represented by a solid homogeneous model with equivalent material properties.

Side walls of TIW were simulated as segments of a conductive thread with the conductivity *G* = 18 kS/m and the diameter *d* = 0.3 mm. The diameter of the thread is given by its construction and material composition. At frequencies above 10 GHz, the distance between stitches should be smaller than the diameter of the thread in the case of TIW antennas.

The screen-printed conductive layers were represented by a perfect electric conductor (PEC). The thickness of screen-printed conductive layers was 0.06 mm.

Figure 2 shows simulated and measured frequency responses of reflection coefficient at the input of the 5.8 GHz antenna. In simulations, an iron-on foil below screen-printed layers was not considered. Therefore, the resonance frequency of the antenna is shifted to lower frequencies. The measured data show that the antenna has the impedance bandwidth 674 MHz and the axial ratio (AR < 3 dB) bandwidth 150 MHz; the minimum value of AR is 1.27 dB at 5.45 GHz.

The left-handed circular polarization (LHCP) radiation patterns are shown in Figure 3 (XZ plane) and Figure 4 (YZ plane) in the span 240°. The measured width of the main lobe is 86° in the XZ plane and 94° in the YZ plane. The measured gain of the antenna is 5.32 dBi. The elevation of the maximum radiation is −7° in the XZ plane and +3° in the YZ plane.

For the 24 GHz ISM band, conductive surfaces of the antennas were manufactured from a standard copper foil and by screen printing. In both the cases, conductive materials were placed on the DIGIFLEX MASTER foil. Using these two technologies, manufacturing intolerances can be compared, and properties of the antennas verified.

The precise SMA connector with the maximum operation frequency 24.5 GHz was used for feeding; the connector was attached to the textile substrate by a conductive glue. Both versions of the antenna were manufactured considering the same source data, and basic parameters were measured.

Using the vector analyzer RS ZVA110, frequency responses of reflection coefficient at the input of antennas were measured (see Figure 5):The antenna was designed for central frequency 24 GHz and simulated in the frequency range of 23 to 25 GHz.Response of the copper foil antenna shows reflections lower than −10 dB with two modes of resonance at frequencies 22.4 GHz and 24.8 GHz. The value of |S11| varies from −12.15 dB to −13.08 dB in the band of interest.The minimum of reflection coefficient of the screen-printed antenna is shifted to the frequency 22.7 GHz, and the response differs both from the copper antenna and the simulated one. The measured impedance bandwidth of the printed antenna is 3.02 GHz.

Numerical experiments indicated that frequency shifts are caused by manufacturing inaccuracies. When producing additional antenna prototypes, frequency responses were shifted within the range of 22 to 23 GHz for each specimen differently.

For the 24 GHz ISM band, conductive surfaces of the antennas were manufactured from a standard copper foil and by screen printing. In both the cases, conductive materials were placed on the DIGIFLEX MASTER foil. Using these two technologies, manufacturing intolerances can be compared, and properties of the antennas can be verified.

Radiations patterns were measured in the XZ plane and YZ plane (see Figure 6 and Figure 7). The measurement was limited to 180° (−90°; +90°) due to good precision achieved in this span. The measured beam width of the copper antenna was 56° in the XZ plane and 84° in the YZ plane. For the screen-printed antenna, the width of the main lobe was 79° in the YZ plane and 95° in the XZ plane. The radiation patterns were measured in the minimum of the axial ratio (22.25 GHz). The gain was measured in comparison with the reference antenna. The gain was 4.3 dBi for the copper antenna and 4.1 dBi for the screen-printed antenna.

Obviously, the simulated gain is higher than the gain of the copper-foil antenna and the screen-printed one. The gain decrease is caused by losses in screen-printed and self-adhesive copper surfaces, in polyester yarns and in a glued connector. Moreover, the mechanical stability and manufacturing precision of a textile substrate is much lower compared to conventional microwave substrates.

Frequency response of the axial ratio of the antenna for the front direction (0°; 0°) is depicted in Figure 8:The antenna was designed for the central frequency 24 GHz and simulated in the frequency range from 23 to 25 GHz. At minimum, the axial ratio (AR) reached the value 1 dB.In case of the copper foil antenna, the AR minimum appears at 23.90 GHz. Since the AR magnitude is higher than 3 dB, the antenna radiates the elliptically polarized wave.In case of the screen-printed antenna, the AR minimum is shifted to the frequency 22.25 GHz. The minimum magnitude of the AR is 1.06 dB, and the AR bandwidth is 1.2 GHz.

Similar to impedance characteristics, frequency shifts are caused by manufacturing inaccuracies. Performing additional numerical experiments and producing additional antenna prototypes, frequency shifted were observed for each specimen differently.

## 4. Conclusions

In the paper, a novel circularly polarized slot antenna was presented. The antenna was designed for full integration into a 3D knitted fabric. Conductive surfaces were screen-printed and the side walls of a textile-integrated feeder were sewed.

The first version of the antenna was designed for the 5.8 GHz ISM band. Simulated and measured results (Figure 2, Figure 3 and Figure 4) were in a reasonable agreement. The antenna showed the impedance bandwidth as 674 MHz and the axial ratio bandwidth as 150 MHz.

The measured gain of the antenna was 5.32 dBi. At 5.8 GHz, the gain of the presented textile-integrated antenna was 4 dB lower than the gain of an advanced patch on a conventional microwave substrate [26]. The gain of textile antennas is usually lower than conventional equivalents [17,19,23]. The decrease is caused by losses in screen-printed surfaces (versus copper), polyester yarns (versus ceramics) and a glued connector (versus a soldered one). Moreover, the mechanical stability and manufacturing precision of a textile substrate is much lower compared to conventional microwave substrates.

The negative influence of textile-related technologies to the gain of an antenna can be observed at 5.8 GHz already. If a lower gain is acceptable, operation frequency can be increased and an influence on other antenna parameters can be observed.

In order to investigate the frequency limits of the textile-integrated implementation, two versions of 24 GHz ISM antennas were developed—the reference copper one and the screen-printed one. Both the antennas were measured and compared with simulations (Figure 5, Figure 6, Figure 7 and Figure 8).

The impedance bandwidth of the simulated antenna was from 23 to 25 GHz. The textile-integrated copper antenna was matched since 21.7 GHz, and the screen-printed one showed the bandwidth from 21.0 to 23.7 GHz. The screen-printed prototype was manufactured several times, but a better match was never achieved. The microscopic view of the slot (Figure 9) reveals that the shift is caused by the limited accuracy of the printing technology.

Similarly, the axial ratio (AR) is shifted for the screen-printed antenna; the textile-integrated copper antenna shows the AR minimum at 24 GHz but the AR value is about 5 dB.

Due to the limited accuracy of screen printing, reduced mechanical stability of the textile substrate, and limited repeatability and accuracy of the production of textile substrates, the operational frequency of 24 GHz is beyond the upper frequency limit as shown in the presented experimental investigation:The realized gain decreases due to losses and manufacturing inaccuracies;The axial ratio is either higher (about 5 dB for the copper foil) or frequency shifted (about 22 GHz for screen printing).Impedance characteristics is shifted towards lower frequencies both for the copper foil and screen printing.

In order to break the frequency limit, mechanical stability of the textile substrate has to be increased and accuracy of manufacturing technologies has to be improved. On the other hand, the study did not consider deformations of the antenna and further influences associated with practical use.

Currently, the research of textile circularly polarized antennas dominantly covers flexible and transparent antennas for reliable wearable applications [27], conformal textile antenna arrays for wearable devices [28], UWB antennas for body-centric communication [19], and multiple-input multiple-output one [29]. In all those emerging concepts, the fundamental principles presented in this paper can be applied to move the operation frequency of textile-integrated radiators higher.

## Figures and Tables

**Figure 1 sensors-22-02707-f001:**
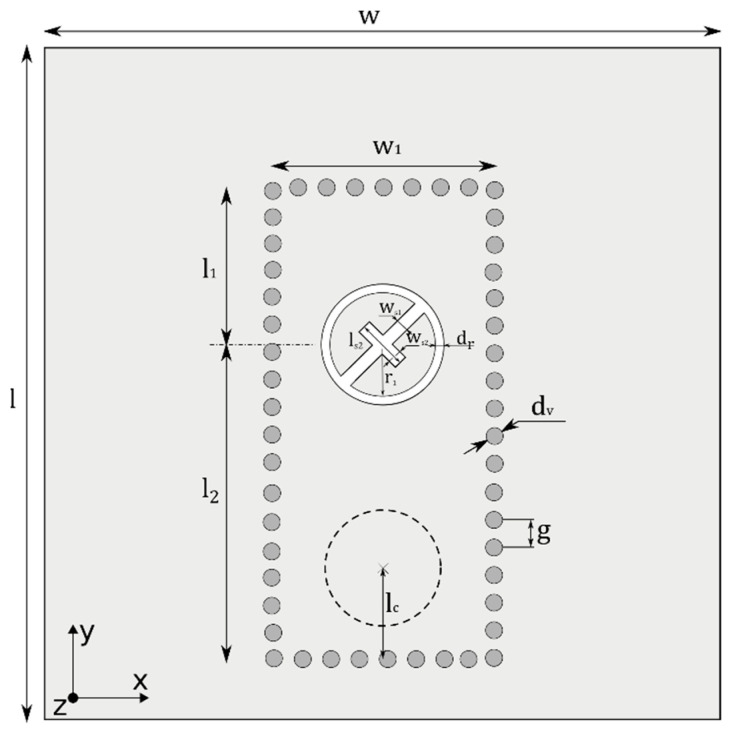
Layout of the designed textile-integrated slot antenna.

**Figure 2 sensors-22-02707-f002:**
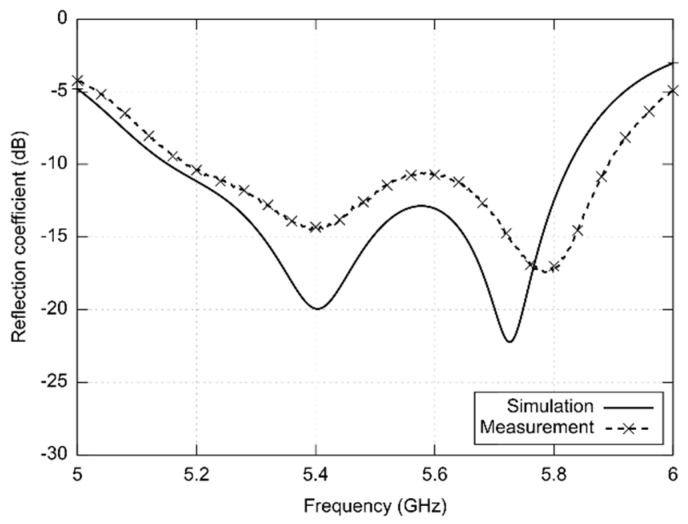
Simulated and measured frequency responses of reflection coefficient at the input of the antenna designed for the 5.8 GHz ISM band.

**Figure 3 sensors-22-02707-f003:**
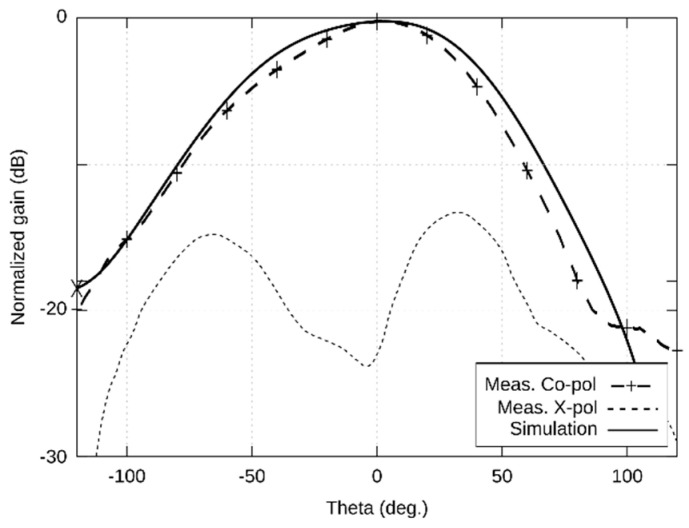
Simulated and measured radiation patterns (XZ plane) of the antenna designed for the 5.8 GHz ISM band.

**Figure 4 sensors-22-02707-f004:**
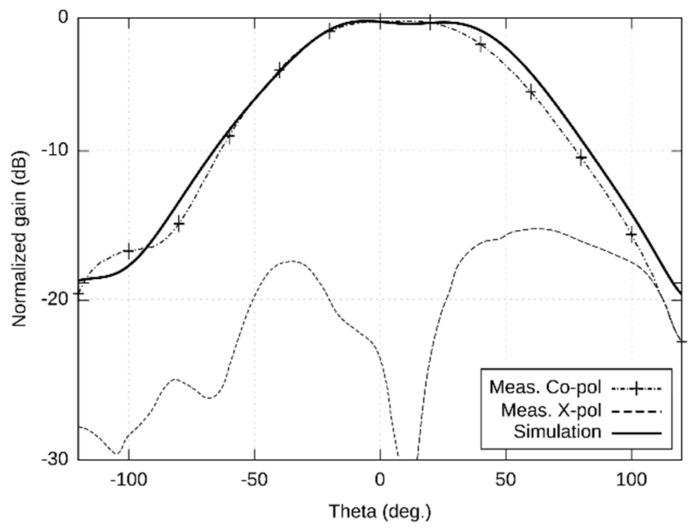
Simulated and measured radiation patterns (YZ plane) of the antenna designed for the 5.8 GHz ISM band.

**Figure 5 sensors-22-02707-f005:**
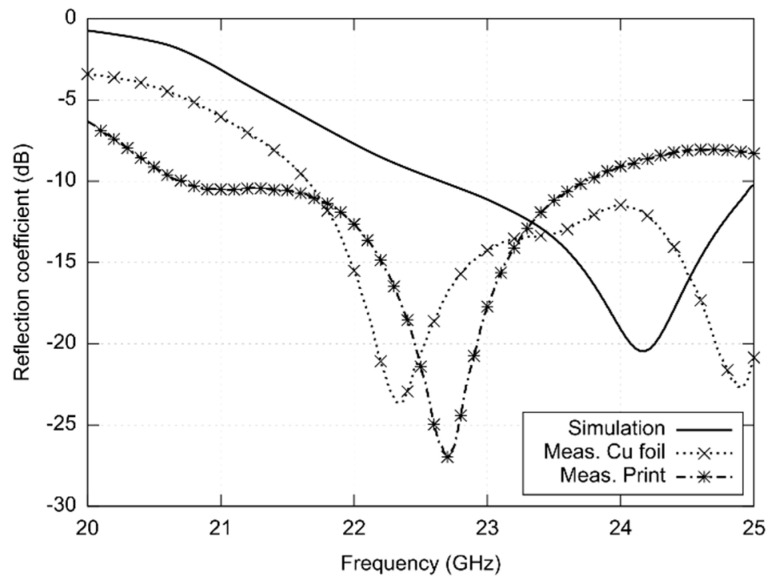
Simulated and measured frequency responses of reflection coefficients at the inputs of antennas designed for the 24 GHz ISM band.

**Figure 6 sensors-22-02707-f006:**
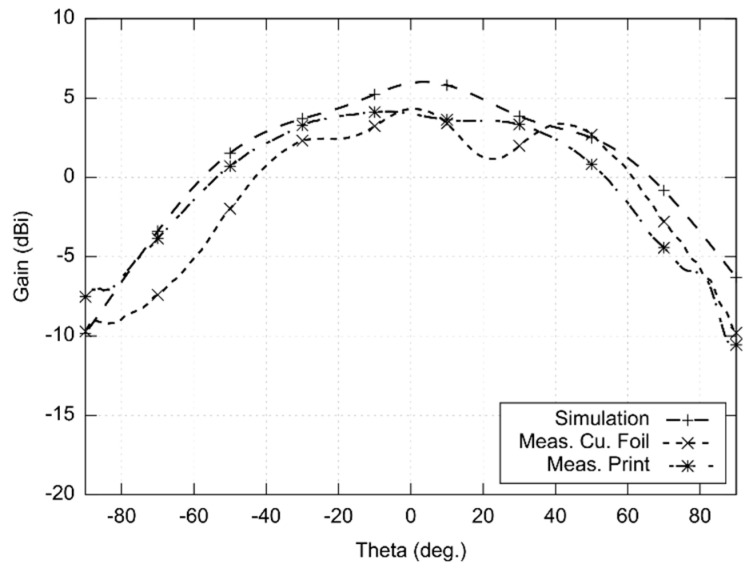
Simulated and measured radiation patterns (XZ plane) of the antenna designed for the 24 GHz ISM band (simulated and measured at 22.25 GHz).

**Figure 7 sensors-22-02707-f007:**
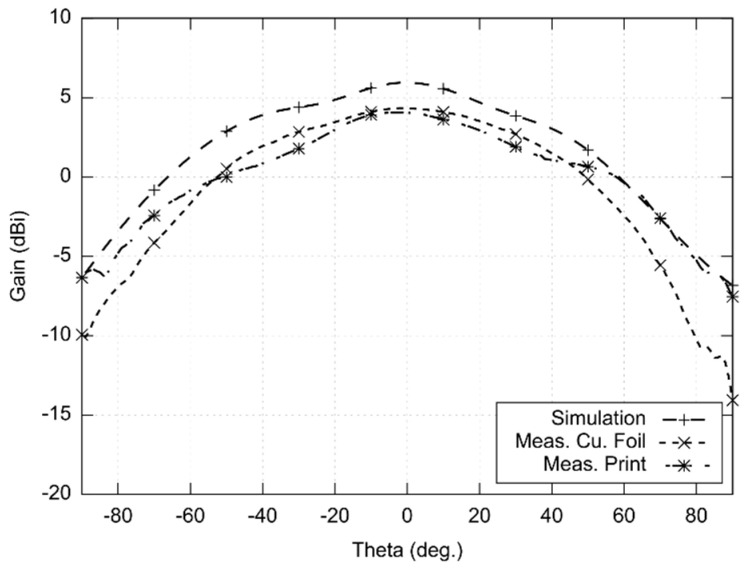
Simulated and measured radiation patterns (YZ plane) of the antenna designed for the 24 GHz ISM band (simulated and measured at 22.25 GHz).

**Figure 8 sensors-22-02707-f008:**
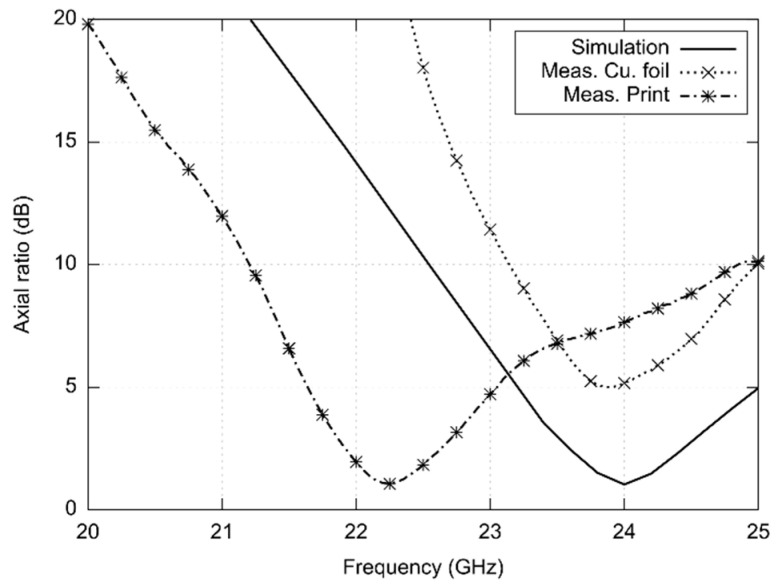
Simulated and measured frequency response of the axial ratio of antennas designed for the 24 GHz ISM band.

**Figure 9 sensors-22-02707-f009:**
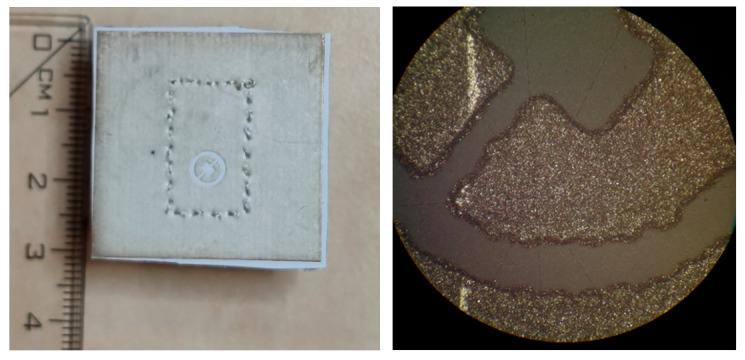
Photograph of the screen-printed antenna (**left**); microscopic view of the slot (**right**).

**Table 1 sensors-22-02707-t001:** Parameters of selected 3D fabrics.

Textile	h (mm)	ε_r_ (-)	tan δ (-)
3D041	3.4	1.22	0.0021
3D097	2.6	1.22	0.0019

**Table 2 sensors-22-02707-t002:** Dimensions of designed antennas.

	** *l* _1_ **	** *l* _2_ **	** *d_v_* **	** *g* **	** *l_c_* **	** *w_s_* _1_ **
5.8 GHz	25.80	55.70	0.30	1.20	22.60	2.50
24 GHz	5.86	11.30	0.30	1.20	3.23	0.42
	** *w_s_* _2_ **	** *l_s_* _2_ **	** *r* _1_ **	** *d_r_* **	** *w* _1_ **	
5.8 GHz	2.70	12.30	9.80	2.50	40.82	
24 GHz	1.28	1.79	1.76	0.65	10.20	

## Data Availability

All relevant data are included in the manuscript.

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
