# Peer review of "Slot Antennas Integrated into 3D Knitted Fabrics: 5.8 GHz and 24 GHz ISM Bands"

_sensors, 2022, doi:10.3390/s22072707_

Round 1

Reviewer 1 Report

  • Is a gain of 5.4 dBi suitable for 24 GHz operation?
  • Why 24 GHz represents the upper limit? Have the authors tried a higher frequency and didn’t achieve the expected results? In fact, the considerable difference between experimental and simulated results demonstrates that the practical upper limit may be well below 24 GHz.
  • In Figure 1, what does the dotted circle represents and what does L stands for?
  • Is a fabrication accuracy of 10% acceptable to design reliable devices? What is the impact of this on sensitive parameters such as the axial ratio of the CP radiation?
  • The parameter g and dv are the same at 5.8 GHz and 24 GHz. How will this impact the performance at the two different frequencies?
  • Figure 2, it would be clearer if a solid and dashed lines are used to define the measured and simulated results.
  • Figures 3 and 4, how you know this an LHCP and not RHCP? It is a common practice to show the EL and ER There is no need for such a large scale as the minimum value is ~-25 dB while the scale starts at -50 dB.
  • Figure 5, the used graph lines are confusing as they look similar. In addition, there is a considerable difference between measurements and simulations. Why are the simulated results presented over a narrower frequency range?  What is the simulated bandwidth?
  • Please comment on the difference between the measured and simulated peak gains of Figures 6 and 7.
  • The minimum AR point is 22.25 GHz for one design, and it is different at the other two designs. Are all the radiation patterns presented at 22.25 GHz or at the minimum AR of each design?
  • Figure 8, comments are needed with respect to the considerable difference between simulated and measured results.

Author Response

See the enclosed file, please.

Reviewer 2 Report

In this paper two different antenna designed at two different frequencies 5.,8 GHz and 24 GHz. The slotted is not shown properly in this design. Need to show the slotted antenna at the two different frequencies

Need to explain between TIW ans SIW. What equation that has been used in this antenna design is it the same equation for SIW and how do you manufacture the hole especially for high frequencies 24 GHz. Need to explain in the methodology

Why there is aword of that in the starting point of your sentences and please replec with a good word.

The photo is not cleared in this manuscript need to have abetter photo and label all the necessary from this photo.

Please include teh simulation result of field sitribution and the radiation pattern of the antenna as well especially at 24 GHz

Detailing the design of your slotted antenna and TIW

Need to add the latest refrences as well such as in 2018,2019,2020 and 2021

Author Response

See the enclosed file, please.

Round 2

Reviewer 1 Report

The authors failed to address to provide satisfactory responses. 

With respect to the following comments that I made in the original review  

Is a fabrication accuracy of 10% acceptable to design reliable devices? What is the impact of this on sensitive parameters such as the axial ratio of the CP radiation?  

  • The authors didnt comment on the impact of such error on the axial ratio.  In addition, 10% fabrication accuracy cannot be tolerated and need to be reduced considerably.

The parameters g and dv are the same at 5.8 GHz and 24 GHz. How will this impact the performance at the two different frequencies?

  • The authors responded with a comment that is totally irrelevant to my point.  

Figure 2, it would be clearer if solid and dashed lines are used to define the measured and simulated results

  • The authors' response "Modification of charts is timedemanding " is totally unacceptable as and the graphs are confusing and unacceptable in the current form.    

Figures 3 and 4, how you know this an LHCP and not RHCP? It is a common practice to show the EL and ER There is no need for such a large scale as the minimum value is ~-25 dB while the scale starts at -50 dB.

  • The authors' response is just a general comment to avoid extra work.  

Figure 5, the used graph lines are confusing as they look similar. In addition, there is a considerable difference between measurements and simulations. Why are the simulated results presented over a narrower frequency range? What is the simulated bandwidth?  

  • No specific answers has been given to my questions  

The same is true for other comments.  

The authors need to take the reviewers' comments seriously. 

The difference between measured and simulated results need to be reduced.

Author Response

See the enclosed fiel, please.

Round 3

Reviewer 1 Report

The authors have addressed my main concerns.